# Just Enough Thinking: Efficient Reasoning with Adaptive Length Penalties Reinforcement Learning

## Abstract

Large reasoning models (LRMs) achieve higher performance on challenging reasoning tasks by generating more tokens at inference time, but this verbosity often wastes computation on easy problems. Existing solutions—supervised fine-tuning on shorter traces, user-controlled budgets, or RL with uniform penalties—either require data curation, manual configuration, or treat all problems alike regardless of difficulty. We introduce Adaptive Length Penalty (ALP), a reinforcement-learning objective tailoring generation length to per-prompt solve rate. During training, ALP monitors each prompt's difficulty measured using online solve rate through multiple rollouts and adds a differentiable penalty whose magnitude scales inversely with that rate, so easy prompts incur a high cost for extra tokens while hard prompts remain unhindered. Post-training DeepScaleR-1.5B with ALP cuts average token usage by 50% without significantly dropping performance. Relative to fixed-budget and uniform-penalty baselines, ALP redistributes its reduced budget more intelligently—cutting compute on easy prompts and reallocating saved tokens to difficult ones—delivering higher accuracy on the hardest problems with higher cost.

## 1 Introduction

Large reasoning models (LRMs) achieve remarkable gains on complex reasoning benchmarks by expending more inference-time compute, with longer chain-of-thought traces often yielding higher performance (Jaech et al., 2024). However, this verbosity comes at a steep price: extended generations inflate compute and memory requirements, increase latency particularly for simple queries. LRMs frequently "overthink" trivial prompts—for example, `DeepSeek-R1` and `Qwen-QwQ32B` generate over 10,000 tokens to answer "What is 2 + 3?" by retracing the same initial reasoning steps—producing cheap but unproductive reasoning trails (Wang et al., 2025). This is likely due to these reasoning models learn to distill the extended search process such that it continues to verify its own answers and backtracks to previous states to start over. A likely cause of this is the lack of regularization during posting training(Xiang et al., 2025). Hence, enabling reasoning models to adaptively allocate inference-time compute is crucial in the era of scaling inference compute.

The challenge is clear: how can LRMs allocate computation adaptively, using minimal tokens for simple problems while preserving extended reasoning for genuinely difficult ones? Current approaches fall into three categories, each with limitations. Supervised fine-tuning on curated shorter traces (Chen et al., 2025; Pouransari et al., 2024) requires expensive data curation and may compromise reasoning quality. User-controlled budgets—whether enforced through early stopping during inference (Muennighoff et al., 2025) or specified via prompts (Aggarwal & Welleck, 2025)—demand manual configuration for each use case. Reinforcement learning (RL) approaches that apply penalties only to correct solutions (Team et al., 2025; Arora & Zanette, 2025) or use predetermined length cutoffs during training (Hou et al., 2025) reduce verbosity but treat all problems uniformly regardless of their inherent difficulty, despite growing evidence that performance and compute efficiency can be improved by conditioning on accurate difficulty estimates (Truong et al., 2025).

We introduce **Adaptive Length Penalty (ALP)**, a simple yet powerful approach that teaches models to recognize problem difficulty and adapt their reasoning length accordingly. The key insight is that

difficulty can be estimated online during training through empirical solve rates—problems consistently solved across multiple attempts are likely easy, while those with low solve rates are genuinely challenging. ALP leverages this signal to apply differential length penalties: easy problems receive strong penalties for excess tokens, while difficult problems can reason extensively with minimal penalty. Crucially, it works naturally with any RL algorithm that relies on group-based advantage estimation, including the widely-used GRPO (Shao et al., 2024) as well as RLOO and Reinforce++ (Ahmadian et al., 2024), incurring no additional computational cost. Applied to DeepScaleR-1.5B, ALP achieves remarkable efficiency gains—reducing average token usage by over 50% while maintaining accuracy. More importantly, our analysis reveals that ALP learns sophisticated adaptation strategies. Pareto efficiency analysis shows ALP uses only 21% of its token budget on the easiest 50% of problems, compared to 50% for fixed-length baselines. This creates a computational surplus that ALP strategically deploys on challenging problems, achieving 5.35× more tokens on hard versus easy problems. These learned strategies prove robust across varying problem distributions, from gradual difficulty increases to extreme scenarios where 60% of problems are competition-level.

Our contributions are:

- We introduce **Adaptive Length Penalty (ALP)**, an RL objective that scales length penalties inversely with online difficulty estimates, teaching models to internalize problem difficulty without manual configuration or inference-time overhead.

- We demonstrate that ALP achieves superior efficiency-performance trade-offs across multiple benchmarks, significantly reducing token usage while maintaining accuracy, and outperforming existing length-control methods.

- We provide a comprehensive analysis showing how ALP learns to redistribute computation from easy to hard problems, maintaining robust performance across diverse difficulty distributions where fixed strategies prove inefficient.

## 2 RELATED WORK

**Inference-Time Scaling** Inference-time scaling, i.e., increasing the number of tokens LLMs use, is an effective way of increasing model performance (Wu et al., 2024). Many prior works demonstrate effective performance improvement by allowing LLMs to perform various search algorithms (Yao et al., 2023; Xie et al., 2024). Snell et al. (2024) showed that scaling inference-time compute can be more effective than the scaling model size, and particularly scaling in-context generation outperforms parallel sampling based methods. Deepseek-r1's DeepSeek-AI et al. (2025) model performance improves greatly with the length of the responses over training. However, this results in a tradeoff where reasoning models spend significantly more tokens overall, even for simpler problems. Chen et al. (2025) finds that the model will keep generating after producing an initial correct solution and that the diversity of reasoning strategies decreases over the trace.

**Length Control in Reasoning Models.** Various approaches have been proposed to control generation length in reasoning models. At the reward design level, Xiang et al. (2025) proposes using discount factors on step-level rewards to balance length and quality, though most current RL frameworks lack step-level reward support. Several works apply length penalties only to correct solutions during RL training (Team et al., 2025; Arora & Zanette, 2025), reducing average length while maintaining performance on successful traces. Muennighoff et al. (2025) take a different approach, using special tokens to trigger early answering, though this prevents models from learning efficient reasoning compression. User-controlled approaches have also gained traction. Aggarwal & Welleck (2025) train models to follow user-specified token budgets through prompts, either exactly (L1-Exact) or as a maximum (L1-Max), achieving significant efficiency gains while outperforming token-based early stopping. Hou et al. (2025) employ iterative training with progressively stricter length constraints (4k→3k→2k tokens), using clipped rewards to enforce these limits.

While existing methods explore various mechanisms for controlling generation length—fixed budgets, user specifications, early stopping, or progressive constraints—they share a critical limitation: they do not adapt length based on the intrinsic difficulty of each problem instance. These approaches apply uniform policies across all problems, inevitably over-reasoning on simple tasks or under-reasoning on complex ones. Our proposed ALP method directly addresses this gap by introducing a difficulty-conditioned penalty that enables models to learn instance-specific computation allocation.

---

**Algorithm 1** Reinforcement Learning with Adaptive Length Penalty

---

1: **procedure** RL WITH ALP(policy $\pi_\theta$)
2:     **for** each training step **do**
3:         **for** each prompt $q$ **do**
4:             Sample $K$ rollouts $\{y^{(k)}\} \sim \pi_\theta$
5:             Compute $p_{\text{solved}}(q)$ via Eq. equation 1
6:             Compute rewards $\{r(y^{(k)}, q)\}$ via Eq. equation 2
7:             Update $\pi_\theta$ using policy gradient update
8:         **end for**
9:     **end for**
10: **end procedure**

---

Rather than requiring manual configuration or applying blanket constraints, ALP teaches models to internalize problem difficulty and use "just enough" computation for each unique challenge.

## 3 METHOD

We propose ALP as an additional reward term in a standard reinforcement learning framework. Our approach is suited for any RL algorithm as long as it benefits from sampling multiple trajectories for a problem. Here is how it works. Let $q$ be the prompt, $y = (y_1, \ldots, y_n)$ be the chain-of-thought trace generated for $q$, with $N = |y|$. Let $\text{answer}(y)$ denote the extracted final answer and $y^*$ be the ground truth answer, and $K$ be the number of independent rollouts for $q$. Concretely, after drawing $\{y^{(k)}\}_{k=1}^{K}$, we compute the empirical *solve rate* online

$$p_{\text{solved}}(q) \ = \ \frac{1}{K} \sum_{k=1}^{K} \mathbf{1}\big[\text{answer}(y^{(k)}) = y^*\big]. \tag{1}$$

High values of $p_{\text{solved}}(q)$ indicate easy prompts and it receives higher penalty in order to reduce the tokens spent on them. Low values signal harder ones, which leads to a much smaller length penalty term, allowing model to spend more token to solve the $q$. We augment the usual accuracy reward with a length penalty whose *weight* is inversely proportional to $p_{\text{solved}}(q)$, and $\beta > 0$ a global coefficient. For each rollout $(y, q)$, we define the composite reward

$$r(y, q) = \underbrace{\mathbf{1}[\text{answer}(y) = y^*]}_{r_{\text{accuray}}} - \underbrace{\beta N \max\big(p_{\text{solved}}(q), K^{-1}\big)}_{r_{length}}, \tag{2}$$

Here $r_{\text{accuracy}}$ is the reward of the correct answers, $r_{\text{length}}$ charges a per-token cost (scaled by $1/N$), weighted by $\beta$ and by the clipped solve rate. Clips at $1/K$ ensure even unsolved prompts incur some penalty. In order to get an accurate estimate of solve rate, it is advised to use $K >> 1$. Full integration into a generic RL framework is demonstrated in Algorithm 1. Within each training step, the policy needs to sample $K$ rollouts for each prompt. Once sampling finishes, we compute $p_{solved(q)}$, and then evaluate the composition reward. Then the RL loop continues as usual. Because many online RL algorithms, including Grouped Relative Policy Optimization (GRPO) (Shao et al., 2024) and Reinforce++ (Ahmadian et al., 2024) use multiple samples to reduce variance in advantage estimation, ALP obtains solve rate without incurring additional computational cost in these cases.

## 4 EXPERIMENTAL SETUP

**Training and Datasets** In this work, we focus on using math domain to demonstrate the effectiveness of ALP, due to easiness to verify and existing open-source framework. We trained `DeepScaleR-1.5B` on DeepScaleR (Luo et al., 2025), which is a filtered compilation of problems from AIME, AMC problems prior to 2023, Omni-Math (Gao et al., 2024), and Still datasets (Min et al., 2024). We select a base model that already has reasoning capability for improved performance. The context window was set to 16384 during training. We train the base reasoning models

with the objective in Eq 2 for 100 steps with a batch size of 512 using GRPO implemented from VeRL (Sheng et al., 2024). We use `DeepScaleR-1.5B` trained with $\beta = 1e - 7$ for comparison in Section 5.2-5.4. The following sections will refer to this trained model as ALP. Details about hyper-parameters are available in Section A.2.

**Evaluation** We evaluate ALP and other models on math problems with different levels of difficulty, including AIME, for which we combine 2024 and 2025, so the size is 60, MATH-500 (Hendrycks et al., 2021), and OlympiadBench (He et al., 2024). We use Pass@1 as our performance metric, and in order to get reliable estimates, we sample 64 answers per prompt for AIME, 16 answers for MATH-500 and OlympiadBench at `temperature=1` and `top=0.7` with inference budgets at $(512, 1024, 2048, 4096)$ for all models. Because L1 models are trained for length-control through user prompt, we append an additional instruction "Think for maximum $N$ tokens." (for L1-Max) and "Think for $N$ tokens" (for L1-Exact) to the user prompt, $N$ is one of the generation length mentioned above. A list of prompts used for specific models are available in Section A.2.

**Comparison** We compare our ALP-trained model with recent and contemporaneous work in both length reduction and length control. Due to resource constraints, we cannot perform head-to-head comparisons for each method by training with the same RL hyperparameters and base model. We directly use the public checkpoints and ensure the base models all have the same size, which are trained on slightly different datasets, hyperparameters, and steps. Here are some recent and concurrent works we include in our comparison:

- **L1-Exact** - trains `DeepScaleR-1.5B` to follow a precise length specified in user prompt.

- **L1-Max** - is further trained from L1-Exact to keep token usage within a max allowance.

- **ThinkPrune-2K** - a method that trains `DeepSeek-R1-Distill-Qwen-1.5B` to reduce token usage by applying a reduced length-clipped objective. We select the model that was trained to gradually reduce length for three iterations RL training (`4k->3k->2k`). We chose this checkpoint because (Hou et al., 2025) report to be this as one of most efficient model with small degradation.

- **R1-Alpha** - reduces the length by penalizing a response length higher than the average length used for a prompt only for the correct solution. This method uses a $\alpha$ to control the tradeoff between Pass@1 and length. Based on result reported in (Arora & Zanette, 2025), we select $\alpha = 0.2$ as it sufficiently reduce output tokens without too much performance degradation.

## 5 RESULTS AND ANALYSIS

In this section, we address three questions about adaptive length penalties in reasoning models. First, we investigate whether ALP achieves meaningful efficiency gains without sacrificing performance compared to existing approaches (Sections 5.1 - 5.2). Second, we examine how ALP learns to adapt—specifically, whether models develop genuine understanding of problem difficulty and how this manifests in their token allocation strategies (Sections 5.3-5.4). Finally, we explore the implications of compression: how does reducing token usage change the nature of reasoning itself, and are these changes consistent across different models and settings (Section A.3)?

### 5.1 HOW MODELS ACHIEVE EFFICIENCY: PARETO ANALYSIS OF ADAPTIVE COMPUTATION

LRMs can reduce token usage through two fundamentally different strategies: uniform compression, where models use fewer tokens across the board, or adaptive allocation, where models intelligently distribute computation based on problem difficulty. To distinguish between these mechanisms, we analyze how models allocate tokens across problems of varying difficulty using Pareto efficiency curves. We combine all problems from MATH-500, OlympiadBench, and AIME to create a diverse difficulty spectrum, then order them by each model's solve rate from easiest to hardest. For each model, we plot the cumulative percentage of tokens used against the cumulative percentage of problems solved (Figure 1, left). This visualization reveals each model's allocation strategy: convex curves indicate adaptive behavior with minimal tokens on easy problems, followed by increased allocation for harder problems, while linear curves indicate uniform allocation with constant token

usage regardless of difficulty. The area under each curve represents total inefficiency, with lower areas indicating better overall efficiency. To quantify these behaviors, we compute two complementary metrics shown in Figure 1 (right). The adaptation ratio measures the average tokens used on hard problems (bottom 30% by solve rate) divided by average tokens on easy problems (top 30%), where values greater than 1.0 indicate adaptive behavior. The efficiency score, computed as one minus the normalized area under the Pareto curve, captures how well models minimize total computation while maintaining performance.

**ALP learns extreme adaptation without sacrificing efficiency.** The results reveal a striking pattern in how ALP allocates computational resources. ALP uses only 21% of its total tokens to solve the easiest 50% of problems, demonstrating extreme frugality on problems it identifies as simple. This contrasts sharply with L1-Exact, which uses exactly 50% of tokens for 50% of problems—a perfectly diagonal line representing uniform allocation enforced by its fixed-length constraint. This parsimony on easy problems enables ALP to achieve a remarkable 5.35× adaptation ratio, allocating over five times more tokens to problems it deems difficult. Crucially, this aggressive adaptation improves rather than compromises overall efficiency, as evidenced by ALP achieving the highest efficiency score of 0.68 among all methods tested.

**Comparison with existing approaches reveals diverse strategies.** The other methods in our comparison exhibit varying degrees of adaptation that illuminate different approaches to length control. The original DeepScaleR model exhibits natural adaptation (2.41×) even without explicit length penalties, indicating that some difficulty awareness emerges during standard training. However after training with L1 objectives, L1-Exact, constrained to output exactly the prompted length, shows virtually no adaptation with a ratio of 1.01×, confirming that fixed-length constraints prevent difficulty-aware allocation, while L1-Max demonstrates limited adaptation (1.36×) despite having flexibility within its maximum budget, suggesting that user-specified constraints may further inhibit adaptation that already exists in the base model. R1-Alpha and ThinkPrune achieve substantial adaptation ratios of 4.57× and 2.81×, respectively, yet their lower efficiency scores (0.64 and 0.61) suggest these methods may overestimate difficulty or allocate tokens suboptimally compared to ALP's learned policy. ALP achieves both the highest adaptation ratio and the highest efficiency score, revealing a fundamental insight into efficient reasoning, as it has discovered that extreme parsimony on easy problems creates a computational budget surplus that can be strategically deployed on challenging problems without increasing average cost.

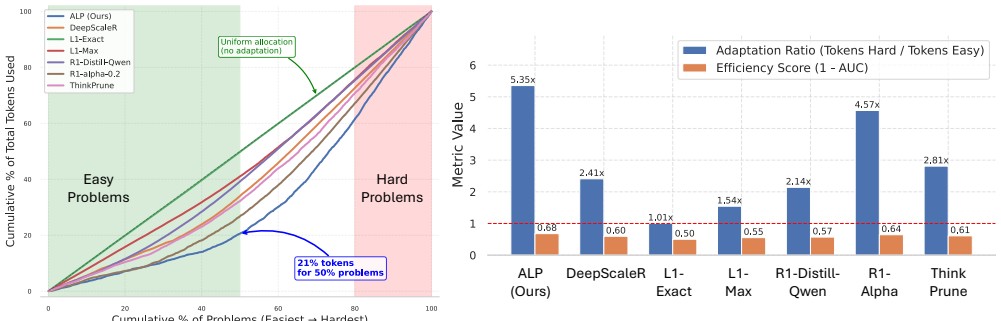

Figure 1: **Pareto efficiency analysis reveals how models distribute computational resources across problems of varying difficulty (inference budget 4096). (Left)** Cumulative token allocation curves for problems ordered from easiest to hardest, aggregated across MATH-500, Olympiad-Bench, and AIME datasets. Shaded regions indicate easy (0-50%) and hard (80-100%) problem ranges. **(Right)** Adaptation ratio is computed as tokens used for hard problems/tokens used for easy problems.

## 5.2 PASS@1-EFFICIENCY TRADEOFFS

**Dramatic improvements over the base model** Compared to the original `DeepScaleR-1.5B`, ALP reduces token usage by approximately 50% on all datasets, particularly at higher compute budget levels, while maintaining performance. Notably, at a budget of 1024 tokens, ALP achieves 40% higher Pass@1 than the base model on easier tasks–math, demonstrating that learned adaptation

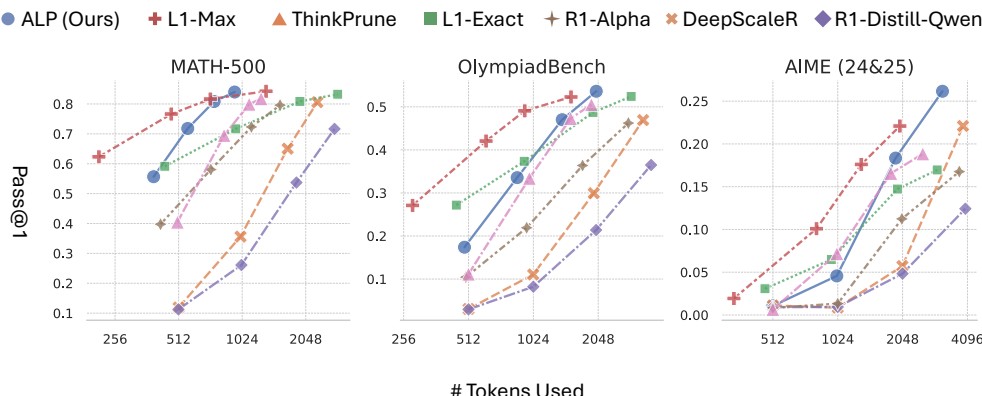

Figure 2: Pass@1 Performance with different inference budgets (512, 1024, 2048, 4096). Inference budget is enforced by setting the max number of generation tokens.

not only reduces average compute but can actually improve performance under constrained budgets by better allocating limited resources. Moreover, we observe that the performance improvement under constrained budgets decreases as the problems become more difficult, suggesting that the models require more tokens to solve them.

**Comparison with other length-reduction methods** Figure 2 presents the trade-off between performance and computational efficiency across three math reasoning benchmarks of increasing difficulty. Across all benchmarks, ALP demonstrates strong performance while using substantially fewer tokens than baselines. The key insight is that ALP's efficiency gains are most pronounced on easier tasks–MATH-500, ALP achieves comparable Pass@1 to all methods while using more than 50% fewer tokens except for L1-Max. This pattern validates our core hypothesis: adaptive computation is most beneficial when problem difficulty varies, allowing models to save substantial compute on routine problems. While L1-Max achieves competitive efficiency on OlympiadBench and AIME, it requires users to specify token budgets a priori—a significant limitation in practice where problem difficulty is unknown. In contrast, ALP automatically adapts its computation without user intervention. L1-Exact, constrained to output exactly the prompted length, cannot optimize the efficiency-performance trade-off and consistently underperforms. ThinkPrune shows reasonable efficiency at higher budgets but struggles at lower budgets ($\leq$2048 tokens), suggesting it reduces length through truncation rather than learned adaptation. R1-Alpha, which applies a fixed length penalty only to correct solutions, shows the poorest efficiency on challenging benchmarks, indicating that uniform penalties fail to capture the relationship between problem difficulty and optimal computation.

## 5.3 ROBUSTNESS TO UNKNOWN DIFFICULTY DISTRIBUTIONS

In practical applications, models encounter problem sets with unknown and varying difficulty distributions. Users cannot—and should not need to—manually adjust computational budgets for each scenario. This necessitates models that can dynamically adapt their reasoning effort based on the actual problems encountered. To evaluate how well different approaches handle varying problem mixtures, we construct controlled experiments mixing MATH-500 (representing standard difficulty) and AIME (representing challenging problems) in different proportions. We examine two complementary scenarios: (1) Gradual shift with N=500 problems, where AIME content ranges from 0% to 12%, simulating typical deployments where most problems are standard difficulty with occasional challenging cases, and (2) Extreme variation with N=100 problems, where AIME content ranges from 0% to 60%, stress-testing model behavior when challenging problems dominate. For each mixture ratio, we measure both accuracy (Pass@1) and computational cost (average tokens per problem), revealing how each method navigates the performance-efficiency trade-off as problem difficulty shifts. Each point on the curves in Figure 3 represents a different MATH/AIME mixture,

with models ideally maintaining high accuracy while adapting their token usage to the problem distribution.

**Adaptive methods excel across all distributions.** Figure 3 reveals markedly different strategies across methods. In the gradual shift scenario (left panel), ALP demonstrates remarkable consistency—maintaining accuracy above 75% even as AIME content increases to 12%, while keeping token usage proportional to difficulty. The curve's moderate slope indicates that ALP successfully identifies which problems require extended reasoning and allocates tokens accordingly. In contrast, L1-Exact maintains constant token usage around 3000 regardless of problem mixture, achieving reasonable accuracy on pure MATH sets but degrading significantly as AIME content increases. This rigid approach wastes computation on easy problems while potentially under-resourcing difficult ones. The extreme variation scenario (Fig 3 right panel) provides a more dramatic illustration of each method's adaptability. When 60% of problems are AIME-level, the performance differences become clear. ALP degrades from 0.89 to 0.52 accuracy while increasing average tokens from approximately 500 to 2200—a measured response to increased difficulty. L1-Max shows limited adaptation within its budget constraints, while methods like R1-Alpha and ThinkPrune achieve moderate efficiency but sacrifice too much accuracy under extreme conditions. Notably, the original DeepScaleR model, despite using the most tokens, fails to match ALP's performance on difficult mixtures, confirming that raw computation without adaptive allocation yields diminishing returns.

These experiments demonstrate that ALP automatically scales its reasoning to match problem requirements. This capability becomes increasingly valuable as language models are deployed in diverse, unpredictable real-world contexts where manual configuration is impractical and fixed strategies inevitably fail.

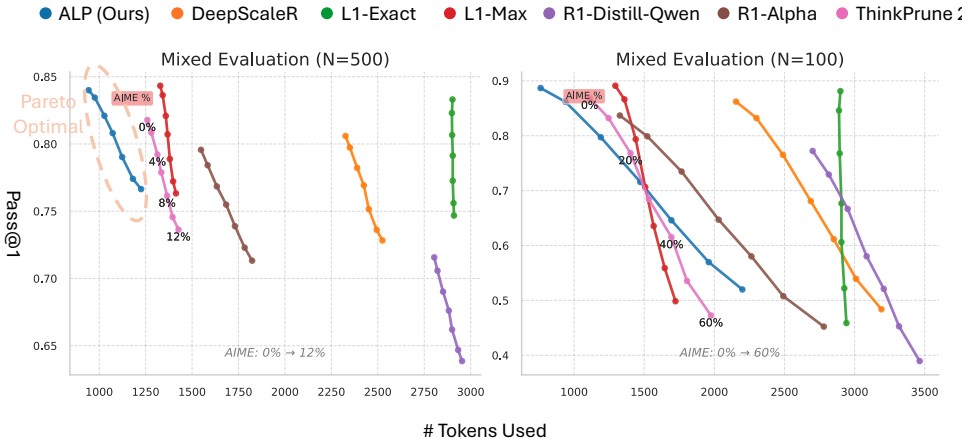

Figure 3: **Performance-efficiency trade-offs under varying problem distributions (inference budget 4096).** Each curve shows model behavior as MATH/AIME mixture changes. **(Left)** N=500 with 0-12% AIME content (typical deployment). **(Right)** N=100 with 0-60% AIME content (stress test). ALP maintains strong performance across all distributions through adaptive token allocation.

## 5.4 HOW MODELS INTERNALIZE PROBLEM DIFFICULTY

While previous sections demonstrated that ALP adapts computation to problem difficulty, a key question remains: Does the model internalize problem difficulty for unseen math problems? Figure 4 reveals that models develop internal difficulty representations through their solve rates—the frequency with which they successfully solve each problem across multiple attempts serves as a proxy for perceived difficulty.

**ALP develops accurate difficulty calibration across datasets.** We analyze token allocation as a function of difficulty, defined as one minus the empirical solve rate (computed using 64 rollouts for AIME, 16 for MATH-500 and OlympiadBench). Higher values indicate problems the model finds more challenging. Across all three datasets, ALP shows a consistent pattern: token usage increases monotonically with difficulty. On MATH-500, ALP uses approximately 500 tokens for

the easiest problems (0.0-0.2 difficulty) but scales up to nearly 3000 tokens for the hardest (0.8-1.0 difficulty)—a 6× increase. This pattern holds on OlympiadBench and AIME, though the absolute token counts are higher, reflecting these datasets' greater inherent difficulty. Importantly, ALP's scaling is smooth and proportional, suggesting it has learned nuanced difficulty assessment rather than binary easy/hard classification.

**Alternative approaches show inefficient or absent adaptation.** In contrast, other methods exhibit problematic patterns. L1-Exact maintains nearly constant token usage (approximately 2900-3000) regardless of difficulty across all datasets, confirming it cannot develop adaptive strategies under fixed-length constraints. More surprisingly, L1-Max shows an inverted U-shape on some datasets, using fewer tokens on the hardest problems—precisely when more computation would be most valuable. DeepScaleR uses excessive tokens uniformly, while R1-Alpha and ThinkPrune show some adaptation but with less dynamic range than ALP. These patterns suggest that explicit adaptive training through difficulty-aware objectives, as implemented in ALP, is necessary for models to develop internal calibration that translates to efficient computation allocation.

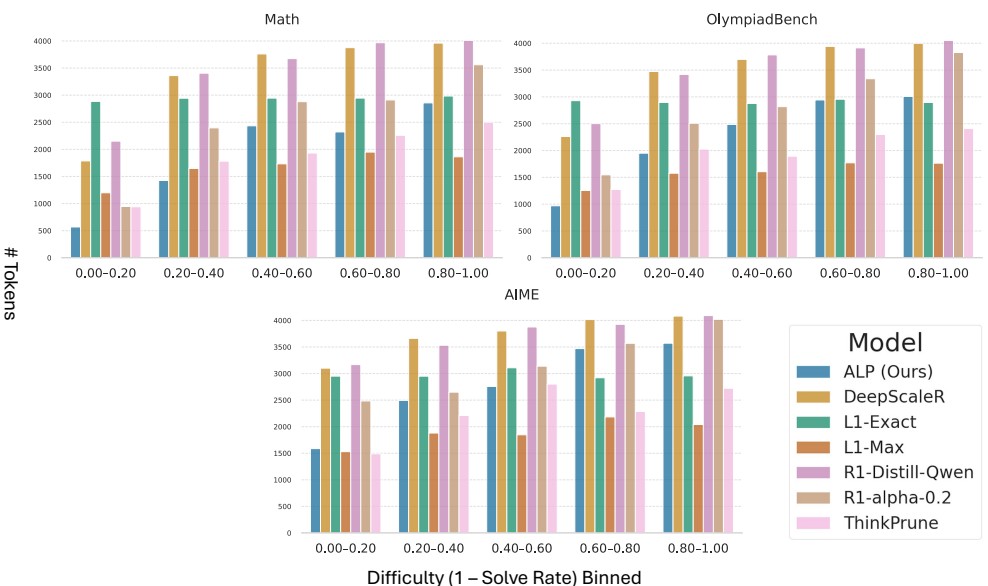

Figure 4: **Token allocation reveals how models internally perceive problem difficulty (inference budget 4096).** Average tokens used versus difficulty (1 - solve rate) across three datasets.

## 5.5 HOW LENGTH REDUCTION CHANGES REASONING BEHAVIOR

While previous sections demonstrated that ALP effectively reduces token usage, a critical question remains: how does this compression affect the model's reasoning process? Do models simply produce more concise versions of the same reasoning, or do they fundamentally alter their problem-solving strategies? To address this, we analyze how specific reasoning behaviors change between the base model and ALP-trained model through systematic keyword and pattern analysis.

We analyzed reasoning behaviors across all three evaluation datasets (MATH-500, OlympiadBench, and AIME, a total of 22624 solutions) by comparing responses from the original DeepScaleR-1.5B model against the ALP-trained variant. For each problem, we extracted and counted specific keyword patterns representing different aspects of math reasoning, e.g. keywords indicating planning, verification or exploration, among other patterns. A list of the behaviors as well as a complete set of the corresponding keywords used to identify each behavior are in Section A.3. We computed average occurrences of these patterns across 500 randomly sampled problems with five rollouts each, providing 2,500 response pairs for analysis.

**Selective compression of reasoning stages.** Figure 5 reveals how ALP selectively compresses different aspects of reasoning. The most dramatic reduction occurs in repetitions, dropping from 269.3

to 125.9 instances on average (52% reduction), confirming that ALP effectively eliminates redundant processing. Exploration behaviors show the second-largest absolute reduction (82.9 to 43.2), suggesting that ALP learns to pursue fewer dead-end solution paths. Interestingly, the compression is not uniform across all reasoning stages. While problem setup (17.1 to 7.9) and verification (7.1 to 2.4) show substantial reductions, planning behaviors are relatively preserved (24 to 10.8, maintaining 46% of the original frequency). This suggests that ALP prioritizes maintaining structured problem-solving approaches while eliminating exploratory wandering and excessive verification. The near-complete elimination of backtracking (34.9 to 13.3) is particularly noteworthy, indicating that the ALP-trained model attempted backtracking less frequently. Conclusion statements show a significant reduction (36.6 to 17), which may be a side effect of the decrease in backtracking. These behavioral changes reveal that ALP does not merely produce shorter versions of the same reasoning—it also alters how models approach problems.

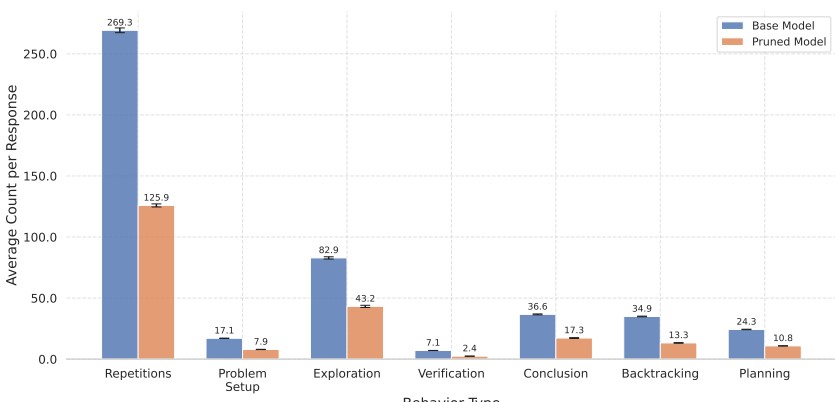

Figure 5: **Reasoning behavior changes reveal selective compression strategies.** Average keyword occurrences per response comparing base DeepScaleR-1.5B (blue) and ALP-trained model (orange) across all evaluation datasets. The average length ratio between ALP and the base model is 49%

## 6 DISCUSSION

We introduce Adaptive Length Penalty (ALP), a principled approach to the fundamental challenge of computational efficiency in reasoning models. By leveraging online difficulty estimation through solve rates, ALP enables models to internalize an understanding of problem complexity without manual configuration. Our analysis reveals that ALP successfully teaches this discrimination, with models learning to compress redundant exploration and repetitive verification while maintaining structured problem-solving approaches. This selective compression mirrors expert reasoning patterns: direct and confident on familiar problems, thorough on challenging ones. Importantly, ALP's learned adaptation strategies prove robust across diverse problem distributions, from standard curricula to competition-level challenges. This robustness suggests that models develop genuine difficulty calibration rather than memorizing specific patterns, making ALP practical for real-world deployment where problem difficulty is unpredictable and manual configuration is infeasible.

## 7 LIMITATIONS

While ALP demonstrates clear efficiency gains, several limitations constrain our conclusions. Most critically, our evaluation focuses exclusively on math reasoning. Whether ALP's benefits extend to other domains remains unclear. Furthermore, ALP optimizes the allocation of existing capabilities rather than enhancing fundamental reasoning abilities. Models become more efficient reasoners but not necessarily better ones. This raises questions about whether efficiency and capability improvements are inherently coupled or can be pursued independently.

ACKNOWLEDGMENTS

Left empty for anonymity.

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

# A APPENDIX

## A.1 MODEL AND HYPERPARAMETER SENSITIVITY

Table 1: Performance and token usage for ALP hyper-param.

| Method | Pass@1 | | | Generation Length | | |
|---|---|---|---|---|---|---|
| | MATH-500 | AIME (24&25) | Olympiad | MATH-500 | AIME (24&25) | Olympiad |
| **DeepScaleR-1.5B** | | | | | | |
| Original Model | 0.81 | 0.22 | 0.47 | 2326 | 3906 | 3309 |
| ALP-8K (beta=1e-7) | 0.8 | 0.24 | 0.51 | 646 | 2254 | 2107 |
| **R1-Distill-Qwen-1.5B** | | | | | | |
| Original Model | 0.72 | 0.12 | 0.36 | 2804 | 4007 | 3606 |
| ALP-8K, $\beta = 1e-7$ | 0.81 | 0.252 | 0.51 | 862 | 3331 | 2107 |
| ALP-4K, $\beta = 1e-7$ | 0.79 | 0.25 | 0.51 | 679 | 2740 | 1631 |
| ALP-4K, $\beta = 1e-8$ | 0.78 | 0.21 | 0.49 | 1063 | 2993 | 2070 |

Pass@1 is estimated using 64 samples per question given 4096 max tokens. ALP-4k and -8K correspond to context window of 4096 and 8192 during training phase. $\beta$ is weighting of ALP.

To assess ALP's robustness and sensitivity, we conducted experiments on two 1.5 B models—DeepScaleR and R1-Distill-Qwen—using identical training settings (100 gradient steps, batch size 512, learning rate $1 \times 10^{-6}$, $K = 16$ rollouts per prompt). First, with a large 8 K-token context window and a single penalty weight-a key hyperparameter which determines the tradeoff between pass@1 rate and token usage-$\beta = 10^{-7}$, both models achieve comparable accuracy on MATH-500, OlympiadBench, and AIME, despite DeepScaleR holding a roughly 10% accuracy advantage in the zero-shot setting (Table 1). Under ALP, R1-Distill-Qwen consumes slightly more tokens on the hardest AIME examples (and marginally more on MATH) while matching performance on OlympiadBench.

Next, keeping model and context size fixed, we swept $\beta$ over $\{10^{-7}, 10^{-8}\}$. The larger penalty ($\beta = 10^{-7}$) drives faster reductions in average token usage, but only yields a modest drop in pass@1: the $\beta = 10^{-8}$ run uses more tokens on MATH and AIME yet underperforms by a small margin. Finally, reducing the context window from 8 K to 4 K tokens further compresses reasoning traces: both base models produce shorter, more efficient solutions with negligible loss in accuracy. The most

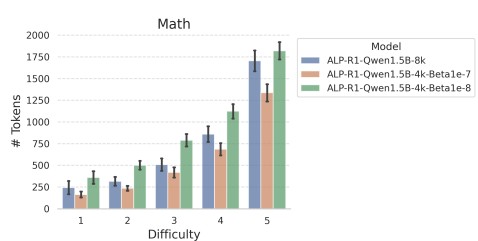

Figure 6: Token usage by MATH-500 difficulty levels.

apparent change observed in Figure 6 is that tokens that signify verification/self-correction become clearly less frequent, as we see the frequency of the "wait" token reduced by two-fold. This aligns with our observation of ALP significantly reducing token usage, as self-verification can lead the model to abandon its current direction completely to either restart or backtrack. We observe minor differences in other key reasoning tokens, but the lack of a clear dropoff in other areas suggests the quality of reasoning might still be preserved despite aggressive pruning.

## A.2 IMPLEMENTATION DETAILS

We provide the exact prompts used for evaluating each model in our experiments. All models use the same math problem as input, with model-specific formatting.

### A.2.1 THINKPRUNE MODELS

For both `DeepSeek-R1-Distill-Qwen-1.5B-thinkprune-iter2k` and `DeepScaleR-1.5B-Preview-thinkprune-iter2k`:

> **System:** A conversation between User and Assistant. The user asks a question, and the Assistant solves it. The assistant first thinks about the reasoning process in the mind and then provides the user with the answer. The reasoning process and answer are enclosed within `<think></think>` and `<answer></answer>` tags, respectively, i.e., `<think>` reasoning process here`</think><answer>` answer here`</answer>`.
> **User:** {problem} Let's think step by step and output the final answer within \boxed{}.

### A.2.2 L1-MAX MODEL

For `l3lab/L1-Qwen-1.5B-Max` with user-specified maximum token budget N:

> **User:** {problem} Let's think step by step and output the final answer within \boxed{}. Think for maximum N tokens.

### A.2.3 L1-EXACT MODEL

For `l3lab/L1-Qwen-1.5B-Exact` with user-specified exact token budget N:

> **User:** {problem} Let's think step by step and output the final answer within \boxed{}. Think for N tokens.

### A.2.4 STANDARD MODELS

For all other models including DeepScaleR, R1-Alpha, DeepSeek-R1-Distilled-Qwen-1.5B and ALP:

> **User:** {problem} Let's think step by step and output the final answer within \boxed{}.

Note: For all models, when a length budget is specified (512, 1024, 2048, or 4096 tokens), generation is truncated at the specified limit. The L1 models require explicit length specification in the prompt, while other models enforce the budget through generation parameters.

**Hyper-parameter for training** Important hyperparameter used for training as in Table 2 for all runs described in our experiments.

**Computational resources** We trained DeepScaleR-1.5B and DeepSeek-R1-Distilled-1.5B on a cluster with 4x8H100 nodes. Training 100 steps with 16324 context window using 4 nodes took

| Parameter | Value |
|---|---|
| Learning rate | $1 \times 10^{-6}$ |
| Batch size | 512 |
| Rollout number | 32 |
| Gradient clipping norm | 1.0 |
| KL Coefficient | 0.001 |

Table 2: Training hyperparameters.

64 hours, 100 steps with 8192 context window using 4 nodes took 34 hours, while training 100 steps with 4096 context window using 1xH100 node took 80 hours.

### A.3 KEYWORDS FOR BEHAVIOR ANALYSIS

The types of reasoning behaviors we tracked using keyword patterns.

- **Repetitions**: Repeated phrases (5+ words), calculations, or questions indicating redundant processing

- **Problem Setup**: Keywords like "given that," "we have," "the problem states"—indicating initial problem understanding

- **Exploration**: Terms such as "what if," "suppose," "let's try"—suggesting hypothesis testing

- **Verification**: Phrases like "check," "verify," "confirm"—showing answer validation

- **Conclusion**: Words including "therefore," "thus," "final answer"—marking solution completion

- **Backtracking**: Indicators like "wait," "actually," "oh no"—revealing self-correction

- **Planning**: Terms such as "first," "then," "next"—demonstrating structured approaches

We used the following keyword patterns to identify different reasoning behaviors in model responses. Each category captures specific aspects of math problem-solving:

#### A.3.1 REASONING STAGES

**Problem Setup:**

```
'let me', 'we have', 'given that', 'the problem
states', 'we need to find', 'we are asked'
```

**Exploration:**

```
'what if', 'suppose', 'consider', 'try', 'perhaps',
'let's see', 'maybe we can', 'alternatively'
```

**Verification:**

```
'check', 'verify', 'confirm', 'make sure',
'double-check', 'let's verify', 'to confirm',
'checking our work'
```

**Conclusion:**

```
'therefore', 'thus', 'so the answer is', 'final
answer', 'in conclusion', 'this gives us', 'the result
is'
```

### A.3.2 BACKTRACKING PATTERNS

**Correction:**

> 'wait', 'actually', 'oh no', 'I made an error', 'let
> me fix', 'that's wrong', 'my mistake', 'correction'

**Reconsideration:**

> 'on second thought', 'alternatively', 'or maybe',
> 'hmm', 'but wait', 'hold on', 'rethinking this'

**Restart:**

> 'let me start over', 'from the beginning', 'scratch
> that', 'starting fresh', 'let's restart', 'back to
> square one'

### A.3.3 PLANNING

> 'first', 'then', 'next', 'finally', 'step by step',
> 'my plan is', 'I'll start with', 'followed by'

**Note:** For the analysis, we counted occurrences of all backtracking subcategories (correction, reconsideration, and restart) together under the unified "Backtracking" category. Repetitions were identified as any phrase of 5 or more words that appeared multiple times within a single response.

