# OpenReview forum: "Just Enough Thinking: Efficient Reasoning with Adaptive Length Penalties Reinforcement Learning"
_ICLR.cc/2026/Conference — Submitted to ICLR 2026_

### Official Review · Reviewer_xGHQ · 2025-10-25

**Soundness:** 3
**Presentation:** 3
**Contribution:** 3
**Rating:** 6
**Confidence:** 4

**Summary:**

It introduces Adaptive Length Penalty (ALP), a reinforcement learning (RL) method that enables large reasoning models (LRMs) to dynamically adjust the length of their reasoning traces based on problem difficulty—using fewer tokens on easy problems and more on hard ones.

A novel RL reward objective that scales token penalties inversely with an online estimate of problem difficulty, measured by empirical solve rate across multiple rollouts.

ALP learns to spend only 21% of tokens on the easiest 50% of problems, freeing up resources for harder ones—achieving a 5.35× higher token allocation ratio (hard vs. easy).

**Strengths:**

1. Compare against strong, recent baselines, including L1-Exact/Max, ThinkPrune, and R1-Alpha, covering supervised, user-controlled, and RL-based length-control strategies.
2. Conduct multiple layers of analysis: Pareto efficiency curves, adaptation ratios, robustness to unknown difficulty mixtures, and fine-grained behavioral analysis of reasoning traces.
3. The work addresses a critical bottleneck in deploying reasoning models at scale: the unsustainable cost of verbose, unadaptive chain-of-thought generation. By enabling models to use “just enough” reasoning, ALP offers a practical path toward more sustainable, responsive, and cost-effective AI systems—especially valuable as models are deployed in real-world settings with unknown or mixed-difficulty inputs.

**Weaknesses:**

1. The model-scale of exp is limited to DeepScaleR-1.5B, which may not be suitable for large-scale model-size such as 7b/32b.
2. The beta in Equation [2] may vary for different model-size, I think the exp should contain more analysis for 7/32b.

**Questions:**

1. This paper analyzes the REASONING BEHAVIOR CHANGES of length reduction, it may change the over/under-thinking, could you show the over/under-thinking metric since the reasoning patterns may not reveal the performance of over/under-thinking.
2. The beta in Equation [2] may vary for different model-size and domains, I am wondering the beta for logical reasoning, STEM and coding rather than math.

---

> ### Author Response · Authors · 2025-12-03
>
> We appreciate the reviewer's recognition of the soundness of our results and effectiveness of our method.
>
> > The model-scale of exp is limited to DeepScaleR-1.5B, which may not be suitable for large-scale model-size such as 7b/32b. Different beta is required for different model sizes.
>
>
> Due to limited compute (1 node of 8 H100s), we were unable to conduct RL experiments beyond 8B scale at bf16 precision. To study the generality of ALP, we have extended our evaluation to **Qwen3-4B** and **Qwen3-8B** (no-thinking) to demonstrate architectural robustness. Using AceReason dataset with, a 16k context window (temp=0.8), we successfully reproduced the "reasoning model" phenomenon, where RL training  eads to significant performance gains alongside a massive increase in response length.
> Our results show that ALP allows us to retain the majority of these performance gains while preventing length explosion. For example, on Qwen3-8B, ALP ($\beta$ 1e-7) and Qwen3-4B ($\beta$ 1e-9) achieves comparable accuracy to GRPO but with much shorter generations. This confirms ALP effectively balances reasoning depth and efficiency across scales.
> Table 1: Qwen3-4B Performance (%)
> | Training | Beta | N | AIME-24 | Length | AIME-25 | Length | AMC | Length | OlympiadBench | Length |
> | :--- | :--- | :--- | :--- | :--- | :--- | :--- | :--- | :--- | :--- | :--- |
> | Untrained | - | - | 22.7 | 5775 | 19.1 | 4499 | 61.8 | 2291 | 46.3 | 2400 |
> | GRPO | - | 16 | 40.4 | 9372 | 33.5 | 9406 | 74.7 | 4479 | 54 | 5079 |
> | ALP |1e-7 | 16 | 30.9 | 6538 | 24.3 | 6088 | 67.8 | 2916 | 50.2 | 3100 |
> | ALP | 1e-8 | 16 | 36.3 | 7692 | 27.3 | 7434 | 70.1 | 3497 | 51.4 | 3742 |
> | ALP | 1e-9| 16 | 37.3 | 7197 | 28.1 | 7090 | 71.5 | 3263 | 52.1 | 3549 |
> | ALP | 1e-9 | 8 | 31.2 | 7375 | 26.1 | 7199 | 68.4 | 3280 | 51 | 3650 |
> | ALP | 1e-9 | 4 | 26.8 | 6012 | 21 | 5065 | 65.5 | 2784 | 48.5 | 2787 |
> Table 2: Qwen3-8B Performance (%)
> | Training | Beta | N | AIME-24 | Length | AIME-25 | Length | AMC | Length | OlympiadBench | Length |
> | :--- | :--- | :--- | :--- | :--- | :--- | :--- | :--- | :--- | :--- | :--- |
> | Untrained | - | - | 25.4 | 5112 | 20.1 | 4076 | 62.1 | 2522 | 46.1 | 2418 |
> | GRPO | - | 16 | 45 | 8730 | 36.8 | 8822 | 77.1 | 4195 | 56.4 | 4487 |
> | ALP | 1e-5| 16 | 36.6 | 5661 | 25.3 | 5698 | 69.4 | 2280 | 52.2 | 2521 |
> | ALP | 1E-6 | 16 | 36.6 | 6029 | 23.2 | 5903 | 68.9 | 2470 | 50.9 | 2866 |
> | ALP | 1e-6 | 8 | 32.8 | 5776 | 23 | 4932 | 67 | 2385 | 51.4 | 2540 |
> | ALP | 1e-6 | 4 | 30.3 | 3711 | 18.5 | 2888 | 63 | 1749 | 47.4 | 1739 |
> | ALP |1e-7 | 16 | 43.4 | 7206 | 30.4 | 7175 | 74.6 | 3185 | 54.5 | 3627 |

---

### Official Review · Reviewer_TiY2 · 2025-10-30

**Soundness:** 3
**Presentation:** 3
**Contribution:** 3
**Rating:** 6
**Confidence:** 3

**Summary:**

This paper addresses the problem of computational inefficiency in Large Reasoning Models (LRMs) caused by overthinking, an excessive reasoning on simple problems. The authors propose Adaptive Length Penalty (ALP), a reinforcement learning objective that dynamically allocates computational effort based on problem difficulty. ALP estimates difficulty using the in-batch solve rate as a real-time proxy and applies a length penalty inversely proportional to it, encouraging concise reasoning for easy problems and longer reasoning for hard ones. The method achieves strong efficiency gains, reducing token usage by 50% without notable accuracy loss.

**Strengths:**

1. ALP’s use of online solve rate as a self-contained difficulty signal is both intuitive and effective. The inverse scaling penalty directly enforces the desired behavior—concise on easy cases, thorough on hard ones—without external classifiers or manual curation.

2. ALP achieves a 50% reduction in average tokens with minimal accuracy degradation. The Pareto efficiency and adaptation ratio analyses (e.g., 5.35× more tokens on hard vs. easy problems) convincingly demonstrate adaptive computation rather than uniform compression.

3. Experiments on mixed-difficulty datasets (e.g., MATH-500 + AIME) show ALP dynamically adjusts computation while maintaining high accuracy, outperforming fixed-length baselines. This supports ALP’s real-world applicability.

4. The paper defines “overthinking” as a concrete inefficiency pattern and positions ALP as a principled fix. The narrative clearly contrasts ALP with prior, static-budget approaches, enhancing clarity and motivation.

**Weaknesses:**

1. The in-batch solve rate, estimated from limited rollouts, may be unstable early in training. Misclassifying an easy problem as “hard” can lead to weak penalties and inefficient behavior.

2. The global penalty coefficient governing the accuracy–conciseness trade-off is fixed without sensitivity testing. It remains unclear how robust ALP is to this choice across models or domains.

3. Evaluation is confined to mathematical reasoning tasks with binary correctness. It is uncertain how ALP’s difficulty signal generalizes to open-ended or subjective tasks lacking a clear notion of “solve rate.”

4. The reasoning-pattern study relies on keyword matching (e.g., “plan,” “verify,” “backtrack”), which is a coarse and gameable proxy. Models may paraphrase such patterns, limiting interpretability of the behavioral findings.

**Questions:**

How stable is the online solve rate estimate during training with limited rollouts? Did you observe inconsistent penalty assignments for the same problem across iterations?

---

> ### Author Response · Authors · 2025-12-03
>
> We thank the reviewer for thoughtful comments and recognize the effectiveness and soundness of our method. We further address the raised concerns and questions below:
> New results:
> Table 1: Qwen3-4B Performance (%)
> | Training | Beta | N | AIME-24 | Length | AIME-25 | Length | AMC | Length | OlympiadBench | Length |
> | :--- | :--- | :--- | :--- | :--- | :--- | :--- | :--- | :--- | :--- | :--- |
> | Untrained | - | - | 22.7 | 5775 | 19.1 | 4499 | 61.8 | 2291 | 46.3 | 2400 |
> | GRPO | - | 16 | 40.4 | 9372 | 33.5 | 9406 | 74.7 | 4479 | 54 | 5079 |
> | ALP |1e-7 | 16 | 30.9 | 6538 | 24.3 | 6088 | 67.8 | 2916 | 50.2 | 3100 |
> | ALP | 1e-8 | 16 | 36.3 | 7692 | 27.3 | 7434 | 70.1 | 3497 | 51.4 | 3742 |
> | ALP | 1e-9| 16 | 37.3 | 7197 | 28.1 | 7090 | 71.5 | 3263 | 52.1 | 3549 |
> | ALP | 1e-9 | 8 | 31.2 | 7375 | 26.1 | 7199 | 68.4 | 3280 | 51 | 3650 |
> | ALP | 1e-9 | 4 | 26.8 | 6012 | 21 | 5065 | 65.5 | 2784 | 48.5 | 2787 |
> Table 2: Qwen3-8B Performance (%)
> | Training | Beta | N | AIME-24 | Length | AIME-25 | Length | AMC | Length | OlympiadBench | Length |
> | :--- | :--- | :--- | :--- | :--- | :--- | :--- | :--- | :--- | :--- | :--- |
> | Untrained | - | - | 25.4 | 5112 | 20.1 | 4076 | 62.1 | 2522 | 46.1 | 2418 |
> | GRPO | - | 16 | 45 | 8730 | 36.8 | 8822 | 77.1 | 4195 | 56.4 | 4487 |
> | ALP | 1e-5| 16 | 36.6 | 5661 | 25.3 | 5698 | 69.4 | 2280 | 52.2 | 2521 |
> | ALP | 1E-6 | 16 | 36.6 | 6029 | 23.2 | 5903 | 68.9 | 2470 | 50.9 | 2866 |
> | ALP | 1e-6 | 8 | 32.8 | 5776 | 23 | 4932 | 67 | 2385 | 51.4 | 2540 |
> | ALP | 1e-6 | 4 | 30.3 | 3711 | 18.5 | 2888 | 63 | 1749 | 47.4 | 1739 |
> | ALP |1e-7 | 16 | 43.4 | 7206 | 30.4 | 7175 | 74.6 | 3185 | 54.5 | 3627 |
>
>
> > The in-batch solve rate, estimated from limited rollouts, may be unstable early in training. Misclassifying an easy problem as “hard” can lead to weak penalties and inefficient behavior.
>
>
> This is a valid concern, but we mitigate it by using a sufficient number of rollouts ($K=16$) to estimate the solve rate during training. Furthermore, since the policy gradient update averages signals over many steps and batches, the noise from individual difficulty estimates washes out. Empirically, we observe stable convergence and consistent performance gains across all benchmarks (approaching training without ALP, see Table 1 and 2), indicating that the online solve rate is a sufficiently robust signal for difficulty differentiation.
>
> > The global penalty coefficient governing the accuracy–conciseness trade-off is fixed without sensitivity testing. It remains unclear how robust ALP is to this choice across models or domains.
>
>
>  We have conducted sensitivity testing on the global penalty coefficient $\beta$. We found the method to be robust: varying $\beta$ allows us to traverse the Pareto frontier, trading off varying degrees of token reduction for accuracy. The relationship is consistent-higher betas yield more aggressive reduction-allowing users to tune the parameter based on their specific latency constraints without breaking the adaptation mechanism.
>
>
> > Evaluation is confined to mathematical reasoning tasks with binary correctness. It is uncertain how ALP’s difficulty signal generalizes to open-ended or subjective tasks lacking a clear notion of “solve rate.”
>
>
> We agree that ALP currently relies on a verifiable "solve rate," which is most naturally defined in math and code. However, the core mechanism-scaling length penalties inversely with a quality metric-is general. For open-ended tasks, the binary solve rate can be replaced by a continuous score from a reward model or an LLM judge.
>
>
> > The reasoning-pattern study relies on keyword matching (e.g., “plan,” “verify,” “backtrack”), which is a coarse and gameable proxy. Models may paraphrase such patterns, limiting interpretability of the behavioral findings.
>
>
> We acknowledge that keyword counting is a proxy. However, the magnitude of the changes we observed-specifically the drastic reduction in repetition and backtracking markers-strongly suggests a fundamental shift in reasoning behavior rather than simple paraphrasing. These behavioral changes track perfectly with the quantitative token reduction, providing strong evidence that the model is genuinely learning to prune redundant thinking steps.

---

### Official Review · Reviewer_aiyV · 2025-11-01

**Soundness:** 3
**Presentation:** 3
**Contribution:** 3
**Rating:** 4
**Confidence:** 4

**Summary:**

The paper deals with efficient reasoning in language models, specifically solving problems with the minimum number of required reasoning tokens. The key idea is a reward function that estimates difficulty using the model's pass rate on a prompt, and penalizes the token usage accordingly.

**Strengths:**

- Timely and relevant problem.
- Simple and intuitive approach based on using pass rates in the reward function.

**Weaknesses:**

- The baselines have several potential confounding factors, such as the training data and hyperparameters. Moreover, the paper chooses two specific model checkpoints from the baselines (ThinkPrune-2K, and α=0.2). These make it difficult to reason about what specifically leads to the performance differences.
   - For example, R1-$\alpha$ performs pretty similarly and also shows adaptivity to problem difficulties (e.g., see Figure 1, Figure 4).
- The paper concludes that "These patterns suggest that explicit adaptive training through difficulty-aware objectives, as implemented in ALP, is necessary for models to develop internal calibration that translates to efficient computation allocation." However, we cannot conclude that the difficulty-aware objectives are *necessary* without trying a wide variety of alternative objectives with other variables held constant.
- The related work mentions that prior work "share a critical limitation: they do not adapt length based on the intrinsic difficulty of each problem instance. These approaches apply uniform policies across all problems, inevitably over-reasoning on simple tasks or under-reasoning on complex ones." It is unclear why a simple length penalty added to the reward could not address this issue. Indeed, the experimental results show that several methods are able to adapt to the intrinsic difficulty of each problem instance. For example, in Figure 4 R1-alpha also shows a correlation between token usage and difficulty.
- Experiments are only done on math reasoning with a 1.5B model. It is unclear whether the results will generalize to other domains and model sizes.

**Questions:**

Please address the points above, including:
- Could the authors perform ablations with alternative choices for the reward design? I would particularly be interested in whether a simple length penalty would work, and methods that resemble the baselines but trained in the same setting.
- The paper claims that existing methods cannot adapt to problem difficulty, but the experimental results (e.g., Figure 4 showing R1-alpha's adaptation) appear to contradict this. Can the authors clarify what specific aspect of adaptation ALP provides that other length penalty methods do not?

---

> ### Author Response · Authors · 2025-12-03
>
> We appreciate the reviewers recognize our method as simple and effective with clear motivation. We address your concerns below:
>
> > confounding factors in baseline including the training data and hyperparameters.
>
>
>  We acknowledge that training all baselines from scratch is computationally prohibitive. To ensure a fair comparison, we utilized public, reproducible checkpoints of efficiency methods (ThinkPrune, L1-Exact, R1-Alpha) paired with base models of the same size. While some baselines (like R1-Alpha) exhibit minor correlations between length and difficulty, our Pareto efficiency analysis demonstrates that ALP achieves a superior trade-off. ALP allocates significantly fewer tokens to easy problems while maintaining robust allocation for hard ones, resulting in a more efficient frontier than baselines that rely on implicit or post-hoc adaptation.
>
>
> > we cannot conclude that the difficulty-aware objectives are necessary without trying a wide variety of alternative objectives with other variables held constant.
>
>
> We argue that explicit difficulty-aware objectives are necessary to maximize efficiency. As shown in our results, models trained with rigid constraints (L1) fail to adapt to difficulty, using constant compute regardless of problem complexity. While base models possess some intrinsic adaptability, ALP significantly amplifies this behavior. Without this explicit signal, models tend to "overthink" easy problems; ALP corrects this by redistributing that compute budget to harder problems, a behavior not observed to the same extent in standard RL or uniform penalty methods.
>
> > It is unclear why a simple length penalty added to the reward could not address this issue. Indeed, the experimental results show that several methods are able to adapt to the intrinsic difficulty of each problem instance. For example, in Figure 4 R1-alpha also shows a correlation between token usage and difficulty.
>
> Response: A simple uniform length penalty forces a compromise: a penalty strong enough to compress easy problems will inevitably hurt performance on hard problems by truncating necessary reasoning. ALP avoids this by dynamically scaling the penalty. Our comparison with R1-Alpha (which uses a penalty relative to the average) shows that ALP outperforms it on challenging benchmarks (AIME, OlympiadBench). This suggests that conditioning the penalty on difficulty (solve rate) is more effective than conditioning it on average length or applying a blanket cost.
>
> > Experiments are only done on math reasoning with a 1.5B model. It is unclear whether the results will generalize to other domains and model sizes.
>
>
> We have extended our evaluation to **Qwen3-4B** and **Qwen3-8B** (no-thinking) to demonstrate architectural robustness. Using AceReason dataset with, a 16k context window (temp=0.8), we successfully reproduced the "reasoning model" phenomenon, where RL training  eads to significant performance gains alongside a massive increase in response length.
> Our results show that ALP allows us to retain the majority of these performance gains while preventing length explosion. For example, on Qwen3-8B, ALP ($\beta$ 1e-7) and Qwen3-4B ($\beta$ 1e-9) achieves comparable accuracy to GRPO but with much shorter generations. This confirms ALP effectively balances reasoning depth and efficiency across scales.
> Table 1: Qwen3-4B Performance (%)
> | Training | Beta | N | AIME-24 | Length | AIME-25 | Length | AMC | Length | OlympiadBench | Length |
> | :--- | :--- | :--- | :--- | :--- | :--- | :--- | :--- | :--- | :--- | :--- |
> | Untrained | - | - | 22.7 | 5775 | 19.1 | 4499 | 61.8 | 2291 | 46.3 | 2400 |
> | GRPO | - | 16 | 40.4 | 9372 | 33.5 | 9406 | 74.7 | 4479 | 54 | 5079 |
> | ALP |1e-7 | 16 | 30.9 | 6538 | 24.3 | 6088 | 67.8 | 2916 | 50.2 | 3100 |
> | ALP | 1e-8 | 16 | 36.3 | 7692 | 27.3 | 7434 | 70.1 | 3497 | 51.4 | 3742 |
> | ALP | 1e-9| 16 | 37.3 | 7197 | 28.1 | 7090 | 71.5 | 3263 | 52.1 | 3549 |
> | ALP | 1e-9 | 8 | 31.2 | 7375 | 26.1 | 7199 | 68.4 | 3280 | 51 | 3650 |
> | ALP | 1e-9 | 4 | 26.8 | 6012 | 21 | 5065 | 65.5 | 2784 | 48.5 | 2787 |
> Table 2: Qwen3-8B Performance (%)
> | Training | Beta | N | AIME-24 | Length | AIME-25 | Length | AMC | Length | OlympiadBench | Length |
> | :--- | :--- | :--- | :--- | :--- | :--- | :--- | :--- | :--- | :--- | :--- |
> | Untrained | - | - | 25.4 | 5112 | 20.1 | 4076 | 62.1 | 2522 | 46.1 | 2418 |
> | GRPO | - | 16 | 45 | 8730 | 36.8 | 8822 | 77.1 | 4195 | 56.4 | 4487 |
> | ALP | 1e-5| 16 | 36.6 | 5661 | 25.3 | 5698 | 69.4 | 2280 | 52.2 | 2521 |
> | ALP | 1E-6 | 16 | 36.6 | 6029 | 23.2 | 5903 | 68.9 | 2470 | 50.9 | 2866 |
> | ALP | 1e-6 | 8 | 32.8 | 5776 | 23 | 4932 | 67 | 2385 | 51.4 | 2540 |
> | ALP | 1e-6 | 4 | 30.3 | 3711 | 18.5 | 2888 | 63 | 1749 | 47.4 | 1739 |
> | ALP |1e-7 | 16 | 43.4 | 7206 | 30.4 | 7175 | 74.6 | 3185 | 54.5 | 3627 |

---

### Official Review · Reviewer_Cb1X · 2025-11-01

**Soundness:** 3
**Presentation:** 3
**Contribution:** 2
**Rating:** 4
**Confidence:** 4

**Summary:**

The paper proposes Adaptive Length Penalty (ALP), a reinforcement-learning objective that scales a per-token penalty by an online estimate of problem difficulty (the solve rate across K rollouts). The goal is to spend “just enough” tokens on easy prompts and allow longer reasoning on hard ones. Experiments on math benchmarks (AIME’24/’25, MATH-500, OlympiadBench) suggest similar or higher Pass@1 with ~50% fewer tokens than several baselines.

**Strengths:**

1. The paper is clearly written.
2. Clear problem motivation and relevance.
3. The method is simple and effective.

**Weaknesses:**

- The core idea (scaling a length penalty by probelm difficulty) lands squarely within a fast-growing body of difficulty-aware, adaptive-thinking work. Recent papers already teach models when (and how much) to think via RL or reward shaping, e.g., AdaptThink learns to switch thinking modes based on problem difficulty, LASER frames efficient reasoning as length-based reward shaping with a target-length step reward. To establish genuine novelty, the paper should articulate a crisp conceptual and algorithmic delta relative to these methods and include matched, head-to-head comparisons under identical training/eval setups. Without such positioning and comparative ablations, the contribution is lack of novelty.

[1] Learn to Reason Efficiently with Adaptive Length-based Reward Shaping

[2] AdaptThink: Reasoning Models Can Learn When to Think

[3] Learning When to Think: Shaping Adaptive Reasoning in R1-Style Models via Multi-Stage RL

- All results are reported on a single model (DeepScaleR-1.5B) and math-centric benchmarks; this makes it hard to assess robustness and transfer. At minimum, the study should replicate on diverse model families (e.g., Qwen and LLaMA) and multiple scales (3B, 7B, etc.) to test whether the learned policy and the compute–accuracy trade-off persist across architectures and capacities. Beyond math, a credible “general reasoning” claim requires evaluation on broad knowledge and hard science suites such as MMLU and GPQA, which probe very different skills than contest math. Including such benchmarks, and reporting both accuracy and efficiency (tokens, latency) at fixed budgets, would materially strengthen external validity.

- Because the length penalty is scaled by the current batch/group solve rate, the learning signal can collapse in scenarios where the model already achieves high accuracy on both easy and hard items: if solve rates saturate (e.g., ~100% for both), the penalty coefficient becomes effectively identical, pushing the policy toward a uniform reduction in length rather than learning to differentiate token allocation by difficulty. This undermines the central claim of adaptive thinking, where models might simply shorten outputs across the board without acquiring a calibrated notion of “hard vs. easy.” The manuscript would be stronger with ablations that (a) remove the solve-rate term (pure length penalty) to test whether adaptation persists, (b) substitute alternative difficulty proxies (e.g., entropy, loss) to demonstrate the necessity of the chosen signal, and (c) include thorough sensitivity sweeps for $\beta$ (penalty weight) and K (rollout count) to probe stability and variance effects; without these, it remains unclear whether the method genuinely learns difficulty awareness versus performing global length pruning.

**Questions:**

See Weaknesses

---

> ### Author Response · Authors · 2025-12-03
>
> We appreciate the reviewers recognize our method as simple and effective with clear motivation. We address your concerns below:
> > Missing  related works:
> [1] Learn to Reason Efficiently with Adaptive Length-based Reward Shaping
> [2] AdaptThink: Reasoning Models Can Learn When to Think
> [3] Learning When to Think: Shaping Adaptive Reasoning in R1-Style Models via Multi-Stage RL
>
>
> We thank the reviewer for highlighting these relevant works. We agree that LASER is a very relevant, concurrent work as it also applies difficulty-aware adaptive length-based reward shaping. However, a key distinction is that LASER relies on pre-computing target lengths using a reference set of data points to clip rewards. In contrast, ALP adds a continuous length penalty based on difficulty estimated online via solve rates during multiple rollouts, without requiring pre-computed targets or reward clipping. This makes ALP simpler while maintaining similar computational efficiency. We will cite LASER as concurrent work. Regarding AdaptThink [2] and Learning When to Think [3], we view these as complementary approaches that focus on the “when to think” (switching) problem, whereas ALP optimizes “how efficiently to think” once reasoning is activated.
>
> > All results are reported on a single model (DeepScaleR-1.5B) and math-centric benchmarks; this makes it hard to assess robustness and transfer.
>
>
> We have extended our evaluation to **Qwen3-4B** and **Qwen3-8B** (no-thinking) to demonstrate architectural robustness. Using a 16k context window (temp=0.8), we successfully reproduced the "reasoning model" phenomenon, where RL training  leads to significant performance gains alongside a massive increase in response length.
> Our results show that ALP allows us to retain the majority of these performance gains while preventing length explosion. For example, on Qwen3-8B, ALP ($\beta$ 1e-7) and Qwen3-4B ($\beta$ 1e-9) achieves comparable accuracy to GRPO but with much shorter generations. This confirms ALP effectively balances reasoning depth and efficiency across scales.
> Table 1: Qwen3-4B Performance (%)
> | Training | Beta | N | AIME-24 | Length | AIME-25 | Length | AMC | Length | OlympiadBench | Length |
> | :--- | :--- | :--- | :--- | :--- | :--- | :--- | :--- | :--- | :--- | :--- |
> | Untrained | - | - | 22.7 | 5775 | 19.1 | 4499 | 61.8 | 2291 | 46.3 | 2400 |
> | GRPO | - | 16 | 40.4 | 9372 | 33.5 | 9406 | 74.7 | 4479 | 54 | 5079 |
> | ALP |1e-7 | 16 | 30.9 | 6538 | 24.3 | 6088 | 67.8 | 2916 | 50.2 | 3100 |
> | ALP | 1e-8 | 16 | 36.3 | 7692 | 27.3 | 7434 | 70.1 | 3497 | 51.4 | 3742 |
> | ALP | 1e-9| 16 | 37.3 | 7197 | 28.1 | 7090 | 71.5 | 3263 | 52.1 | 3549 |
> | ALP | 1e-9 | 8 | 31.2 | 7375 | 26.1 | 7199 | 68.4 | 3280 | 51 | 3650 |
> | ALP | 1e-9 | 4 | 26.8 | 6012 | 21 | 5065 | 65.5 | 2784 | 48.5 | 2787 |
> Table 2: Qwen3-8B Performance (%)
> | Training | Beta | N | AIME-24 | Length | AIME-25 | Length | AMC | Length | OlympiadBench | Length |
> | :--- | :--- | :--- | :--- | :--- | :--- | :--- | :--- | :--- | :--- | :--- |
> | Untrained | - | - | 25.4 | 5112 | 20.1 | 4076 | 62.1 | 2522 | 46.1 | 2418 |
> | GRPO | - | 16 | 45 | 8730 | 36.8 | 8822 | 77.1 | 4195 | 56.4 | 4487 |
> | ALP | 1e-5| 16 | 36.6 | 5661 | 25.3 | 5698 | 69.4 | 2280 | 52.2 | 2521 |
> | ALP | 1E-6 | 16 | 36.6 | 6029 | 23.2 | 5903 | 68.9 | 2470 | 50.9 | 2866 |
> | ALP | 1e-6 | 8 | 32.8 | 5776 | 23 | 4932 | 67 | 2385 | 51.4 | 2540 |
> | ALP | 1e-6 | 4 | 30.3 | 3711 | 18.5 | 2888 | 63 | 1749 | 47.4 | 1739 |
> | ALP |1e-7 | 16 | 43.4 | 7206 | 30.4 | 7175 | 74.6 | 3185 | 54.5 | 3627 |
>
> > if solve rates saturate (e.g., ~100% for both), the penalty coefficient becomes effectively identical, pushing the policy toward a uniform reduction in length rather than learning to differentiate token allocation by difficulty.
>
>
> We understand the reviewer's concern regarding signal saturation; however, our analysis suggests that high solve rates actually provide a distinct and necessary signal rather than a collapsed one. Even when the solve rate approaches saturation for a batch, ALP continues to function effectively, as the individual length differs among rollouts even for the same problem.

---

### Author Response · Authors · 2025-12-03
**General Response: New Experiments on Qwen3-4B/8B and Robustness AnalysisWe thank the reviewers for their insightful and constructive feedback.**

We are encouraged that reviewers found our problem setting "important" (Cb1X, TiY2), our method "novel" and "effective" (aiyV), and our analysis of reasoning patterns "interesting" (xGHQ).
A primary concern shared by reviewers (Cb1X, aiyV, xGHQ) was that our experiments were limited to DeepScaleR-1.5B. To address this, we have extended our training and evaluation to Qwen3-4B and Qwen3-8B. We reproduced the "reasoning explosion" phenomenon using GRPO (where length doubles for significant performance gains) and applied ALP to manage it. We found that ALP effectively curbs length explosion across all tested scales (1.5B, 4B, 8B) and on a new back bone. On Qwen3-8B, ALP ($\beta=1e-7$) achieves 43.4% on AIME-24 (approaching the GRPO baseline of 45.0%) but reduces generation length from 8,730 to 7,206 tokens.Parameter Sensitivity: We observe that optimal $\beta$ scales with model capability; larger models (8B) could sustain higher penalties ($\beta=1e-7$) compared to smaller models (4B, $\beta=1e-9$) while maintaining reasoning accuracy.

We provided updated results in Table 1 and 2 in responses to reviewers and will incorporate these results and the accompanying sensitivity analysis into the final version.

---

### Meta-Review · Area_Chair_3j9D · 2026-01-08

**Summary:**

This submission studies the problem of computational inefficiency in large reasoning models, focusing on the tendency of RL-trained models to over-generate long reasoning traces even for simple problems. The authors propose Adaptive Length Penalty (ALP), an RL objective that applies a per-token penalty scaled by an online estimate of problem difficulty, measured via the in-batch solve rate across multiple rollouts. The intended effect is to encourage concise reasoning on easy problems while preserving longer reasoning for harder ones.

Reviewers consistently agreed that the problem setting is important and timely, and that the paper is clearly written with a method that is simple, intuitive, and empirically effective. Empirical results show substantial reductions in average token usage (often around 50%) with relatively modest degradation in accuracy on math-centric benchmarks. Several reviewers also appreciated the breadth of analyses, including Pareto efficiency curves, adaptation ratios, and qualitative reasoning-pattern studies, which together suggest that the model learns to reallocate computation across difficulty levels rather than uniformly truncating outputs.

However, the decision discussion was driven by substantial concerns regarding novelty, experimental control, and the strength of the paper’s causal claims. Multiple reviewers pointed out that ALP lies squarely within a rapidly growing line of work on adaptive reasoning, difficulty-aware reward shaping, and length control. While the authors argue that ALP differs from prior methods by using an online, continuous solve-rate signal rather than precomputed targets or averaged penalties, reviewers were not fully convinced that this distinction constitutes a clear conceptual or algorithmic advance. The lack of head-to-head comparisons under identical training and evaluation conditions further weakens the case for novelty.

A second major concern was the absence of decisive ablation studies. Reviewers repeatedly questioned whether the observed gains require difficulty-aware scaling at all, or whether similar behavior could be achieved with simpler alternatives such as a uniform length penalty or slightly modified versions of existing baselines trained under the same setup. Although the rebuttal provides qualitative arguments and additional parameter sweeps, it does not include controlled experiments that isolate the specific contribution of the solve-rate term, leaving open the possibility that ALP primarily functions as a form of global length regularization.

The original submission was also criticized for its limited experimental scope, relying on a single 1.5B model and math-only benchmarks. The rebuttal partially addresses this by adding experiments on Qwen3-4B and Qwen3-8B, demonstrating that ALP can curb reasoning-length explosion across architectures and scales. While these results strengthen the robustness story, they do not resolve concerns about generalization beyond math-style tasks with binary correctness, nor do they directly address the novelty and attribution issues.

**Reviewer Concerns:**

**Concerns addressed by the rebuttal:**

Model scale and architectural robustness: New experiments on Qwen3-4B and Qwen3-8B partially address concerns about generalization beyond a single 1.5B model and show that ALP mitigates length explosion across scales.

Hyperparameter sensitivity: Added sweeps over the penalty coefficient clarify how β trades off accuracy and efficiency and how it scales with model size.

Stability of the solve-rate signal: The authors provide a plausible argument and empirical evidence that noise in the online solve-rate estimate does not destabilize training.

**Concerns that remain outstanding:**

Limited novelty relative to prior work: Despite improved positioning in the rebuttal, ALP remains closely aligned with a growing body of difficulty-aware and length-based reward shaping methods (e.g., LASER, AdaptThink, R1-style approaches). The distinction—online continuous scaling versus precomputed or averaged targets—is incremental and not convincingly shown to yield fundamentally new capabilities.

Lack of decisive ablations: The paper still does not include controlled experiments that remove the solve-rate term, replace it with alternative difficulty proxies, or compare against simple uniform length penalties under identical training conditions. As a result, it remains unclear whether the observed gains stem from the specific difficulty-aware design or from general length regularization.

Overstated necessity claims: Several reviewers noted that baseline methods already exhibit nontrivial adaptation to problem difficulty. The paper’s claim that explicit difficulty-aware objectives are necessary for efficient computation is not fully supported by the presented evidence.

Narrow evaluation domain: Even with added models, evaluation remains confined to math-style tasks with binary correctness. Claims about broader “general reasoning” efficiency are therefore speculative.

**Reviewer Scores:**

Reviewer Cb1X: Likely remains at 4 as novelty concerns persist.

Reviewer aiyV: Likely remains 4, as core questions about necessity and ablations are not fully resolved.

Reviewer TiY2: Likely remains at 6, viewing the method as solid but limited in scope.

Reviewer xGHQ: Likely remains at 6, with scale concerns partially addressed but novelty still moderate.

---

### Decision · Program_Chairs · 2026-01-26

Reject